# Design of Diamond Bits Water Passage System and Simulation of Bottom Hole Fluid Are Applied to Seafloor Drill

**Jialiang Wang [1,2,\*], Dilei Qian [1,2], Yang Sun [1,2] and Fenfei Peng [1]**

[1] National Local Joint Engineering Laboratory of Marine Mineral Resources Exploration Equipment and Safety Technology, Hunan University of Science and Technology, Xiangtan 411201, China; hnustqian@163.com (D.Q.); sy1401438776@gmail.com (Y.S.); fenfeipeng@163.com (F.P.)

[2] School of Mechanical Engineering, Hunan University of Science and Technology, Xiangtan 411201, China

\* Correspondence: Jialiangwang2019@163.com; Tel.: +86-13507313232

**Abstract:** The performance of the diamond bit directly affects the drilling efficiency of the seafloor drill. The drill bits used in land drilling are prone to abnormal wear, low coring efficiency, and large sample disturbance in marine exploration. At first, in this paper, the operation and formation characteristics of a seafloor drill are utilized to design a water passage system for bottom-jetting diamond bits based on the multi-objective optimization theory. Additionally, then, fluid dynamics theory and the effects of bit rotation on the flow field at the hole bottom were used to analyze the impact of structural and drilling parameters of the HQ-size bit on the flow field of the waterway system. The linear regression equation of the influence of drilling parameters on the bottom hole velocity field and pressure field is obtained. Finally, a field drilling test of the drill bit was carried out. Considering the effect of the grinding length ratio of the bit on the lopsided wear of the inner and outer diameters, the water passage system parameter design and maximum projection area of the cutting tooth are effective optimization goals to improve the normal service life of the bit. The flow field of the drilling fluid at the hole bottom becomes more turbulent and the efficiency of the carrying cuttings return decreases as the waterway height of the bit increases. The optimal bit rotation speed is 250–400 rpm. When drilling into conventional formations, the pump displacement should be controlled within the range of 50–80 L/min. When drilling into sediment formations, the pump displacement should be controlled within the range of 50–65 L/min. An on-site drilling test verified the rationality of the bit water passage system. This work may enrich the existing theories and designs of the water passage system.

**Keywords:** seafloor drill; water passage system; diamond bit; drilling fluid; fluid simulation

## 1. Introduction

The seafloor is rich in mineral resources such as oil, natural gas, combustible ice, polymetallic ooze, and cobalt-rich manganese nodules [1,2]. Mineral reserves can be evaluated to directly obtain cores from the seafloor by drilling. There are two main sampling methods used for exploring marine mineral resources: large drilling ships and seafloor drills [3]. When using the large drilling ship for operation, the drilling tower is located on the deck of a mother ship. With the cooperation of the drilling tower, the drill pipes are connected to the seafloor one-by-one and the drilling process begins. After each drilling run, the coring barrel filled with the samples is lifted from the hole bottom to the mother ship through a fishing device, then a new core barrel is lowered to begin the subsequent drilling process. When using a seafloor drill, the rig and all core barrels are directly lowered to the seafloor through an armored cable as an operator controls them remotely from the mother ship. After the end of a single drilling footage, the core-filled core barrel is first lifted from the hole bottom and stored in the drilling tool library of the rig before a new core barrel is lowered to start the next drilling process. This cycle is repeated until the drilling operation is complete and the rig is recovered to the mother

ship. The operation mode of the seafloor drill, unlike those of large drilling ships, has more lax requirements for mother ship configurations and can be performed quicker; it also has higher operation efficiency. When drilling within 200 m is carried out within a water depth of 3000 m, the seafloor drill is more advantageous in terms of both performance and cost [4–6].

The performance of the diamond bit directly affects the drilling efficiency of seafloor drills. Usually, the bit used in land drilling can meet the requirements in seabed drilling with shallow drilling depth. However, with the increasing depth of seabed drilling operation, the probability of premature failure caused by an abnormal wear of the bit during drilling is significantly increased. Sedimentary strata mixed with hard rock strata dominate the space within 200 m of the seafloor. When drilling in this type of formation, the seafloor drill can be adaptively switched in real time between the push coring and rotary coring modes according to changes in the formation characteristics [7,8]. Whether the design of the bit water passage system is reasonable or not directly affects the drilling efficiency and service life of the bit. Compared with diamond bits used in conventional land drilling, seafloor drills require a more complicated water passage system design. For instance, the drilling fluid pump frequently opens and closes during the adaptive switching between push coring and rotary coring drilling modes. If the design of the bit water passage system is unreasonable, the probability of premature scrapping due to burnout and abnormal wear increases significantly. Additionally, the flow rate and head of the built-in water pump of the seafloor drill are lower than the mud pump used in a land core drilling rig; they also utilize seawater directly as the borehole drilling medium. The ability of seawater to carry cuttings, the wall protection performance, and the cooling effect on the cutting teeth of the bit are lower than those of the mud drilling medium [9,10]. Further, when drilling in sediment formations, the drilling fluid readily causes severe erosion on the hole wall and core. It is necessary to reasonably control the drilling fluid core surface velocity and return velocity as the bit water passage system is operated [11,12].

Core drilling uses conventional-type or bottom jetting-type diamond bits depending on the differences in water passage systems [13,14]. After passing through the clearance between the bit and the core lifter case, the drilling fluid of the conventional bit returns along the annulus between the hole wall and the outer diameter of the drill pipe. The primary working formation of the seafloor drill comprises sediments, as mentioned above. Selecting a bottom jetting bit prevents direct erosion of most drilling fluids on sediments and improves the coring recovery rate. The water passage system of the bottom jetting bit mainly includes waterways, nozzles, and internal and external water grooves. There are few literature reports on the design method of the bottom jetting bit water passage system. The existing water passage system of the bottom jetting bit mainly refers to the conventional bit design concept or directly based on engineering experience, and the design process usually involves ignoring the effects of the nozzle quantity on the bottom crown area of the bit and the grinding ratio of its inner and outer diameters [15–18]. When the distribution of diamonds in the matrix is uniform and the bit has a rectangular waterway shape, the cutting tooth is fan-shaped. Therefore, the working load of diamonds near the inner diameter edge is significantly higher than that of the diamonds near the outer diameter edge. When drilling in complex formations, the inner diameter of the bit is more prone to wear; this renders the core unable to smoothly enter the core pipe due to its diameter, scrapping the bit in advance [19]. When the ratio of the outer diameter to the inner diameter of the bit is greater than 1.2, the risk of lopsided wear on the bit increases significantly [20]. To satisfy the working conditions of seafloor drills, it is obviously essential to design the bottom jetting bit water passage system.

Based on the basic theory of multi-objective optimization, this paper proposes a water passage system design method that is suitable for bottom jetting bits based on seafloor drill formation and operation characteristics. The proposed method is based on fluid dynamics theory and accounts for the effects of bit rotation on the flow field at the hole bottom. The effects of the structural parameters of the water passage system and the bit drilling

parameters on the flow field at the hole bottom are analyzed. Field drilling tests were conducted to verify the rationality of the scheme. This work may enrich the existing design theory for water passage systems, improve the operational efficiency of seafloor drills in soft and hard interlaced complex formations, and help to satisfy the practical demand for lightweight and miniaturized marine resource exploration equipment.

## 2. Bit Water Passage System Design

### 2.1. Bit Water Passage System Structure

The seafloor drill uses wire-line coring drilling technology, so the cutting teeth of the bit matched with the rig have a relatively thick wall. The bit water passage system was designed with main waterways and assistant waterways to enhance the cooling effect of the drilling fluid on its cutting teeth and to control the grinding ratio of its inner and outer diameters, thus preventing lopsided wear. Nozzles are set in the main waterways and the assistant waterways. The assistant waterways arranged on the cutting teeth of the bit are not completely penetrated along the inner diameter direction, which ensures an effective working area for the cutting teeth and reduces the likelihood of lopsided wear.

Figure 1 shows the bottom crown structure diagram of the bottom jetting bit, where $r_1$ and $r_2$ are the main and assistant nozzles radius, respectively, and $d$ and $D$ are the inner and outer diameter of the bit. For the sake of convenient manufacturing, a rectangular nozzle is adopted. The width of the inner and outer water groove is equal to the width of the waterway. The width of the main waterway $b_1$ is 2 ($r_1$ + 1) and that of the assistant waterway $b_2$ is 2 ($r_2$ + 1).

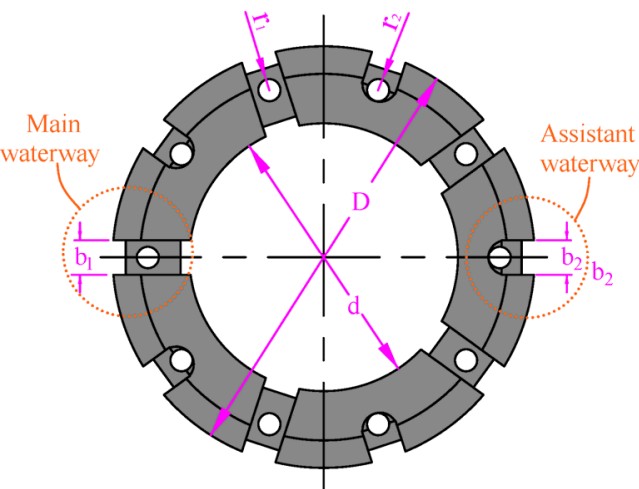

**Figure 1.** Bottom crown structure of bit.

### 2.2. Design Principle and Water Passage System Parameters of Bit

Most of the bottom jetting bit flows return along the outside annulus after flowing through the nozzles; only a small portion of the drilling fluid passes through the water clearance between the core lifter case and the rigid body of the bit. Therefore, the flow rate generated from the water clearance on the flow rate of the outside annulus can be ignored. According to the law of mass conservation, the total flow rate of the drilling fluid flowing through the nozzles is equal to the flow rate flowing through the outside annulus, that is, the product of the average velocity of the cross-section flowing through and the cross-sectional area. This can be expressed as follows [21,22]:

$$V_2 S_T = V_1 \frac{\pi}{4} (D_1{}^2 - D_2{}^2) \tag{1}$$

where $S_T$ is the total water passing area of the nozzles, $D_1$ is the diameter of the hole wall, $D_2$ is the outer diameter of the drill pipe, $V_1$ is the velocity of the drilling fluid along the outside annulus, and $V_2$ is the outlet velocity of the nozzles.

The total water passing area of the nozzles is $S_T$. There are $n$ respective main and assistant nozzles.

$$S_T = n\pi r_1^2 + n\pi r_2^2 \tag{2}$$

The main waterway area is the product of the bit wall thickness $(D - d)/2$ and the main waterway width $b_1$. The area of the assistant waterways is approximately a product of half the wall thickness of the bit $(D - d)/4$ and the width of the assistant waterways $b_2$. Therefore, the ratio of the total area of the waterways to the projected area of the bit crown surface can be calculated as follows:

$$S(r_1, r_2) = \frac{nb_1\frac{D-d}{2} + nb_2\frac{D-d}{4}}{\frac{\pi}{4}(D^2 - d^2)} \tag{3}$$

The minimum $S(r_1, r_2)$ is taken here as the objective function to increase the projected area of the crown of the bit while satisfying the smooth flow of the water passage system and the normal return of carried cuttings. The above equations can be combined accordingly:

$$\begin{cases} S_{\min}(r_1, r_2) = \frac{n(2r_1+2)\frac{D-d}{2} + n(2r_2+2)\frac{D-d}{4}}{\frac{\pi}{4}(D^2 - d^2)} \\ r_1^2 + r_2^2 = \frac{S_T}{n\pi} \\ r_1 - r_2 \geq 0 \\ r_1, r_2 > 0 \end{cases} \tag{4}$$

The ratio of the outer circumference $l_D$ to the inner circumference $l_d$ of the bit can be taken as the grinding ratio $l_k$:

$$l_k = \frac{l_D}{l_d} = \frac{\pi D - 2n(r_1 + 1) - 2n(r_2 + 1)}{\pi d - 2n(r_1 + 1)} \tag{5}$$

The minimum value of $l_k$ is the target under the premise of satisfying the normal working life of the bit while reducing the degree of lopsided wear on its inner and outer diameters.

### 2.3. Calculation of Water Passage System Parameters

The bit used in this study is HQ size, which is often used for geological small-diameter core drilling. The outer diameter of the bit is 96 mm and the inner diameter is 62 mm. The outer diameter of the drill pipe is 89 mm and the hole wall diameter is 97 mm. According to the seafloor core drilling regulations, the upward return velocity range of drilling fluid is 0.75–1.4 m/s and the outlet velocity range of the nozzles is 4–7 m/s [23]. The design value of the return flow velocity is $V_1 = 0.75$ m/s, according to the reverse design principle and under the premise of ensuring the normal return of cuttings; this setting minimizes erosion of the hole wall due to the excessive velocity of the drilling fluid. To prevent repeated grinding of the bit matrix caused by the enrichment of cuttings at the hole bottom, the drilling fluid needs sufficient flow velocity as it exits the nozzles. The nozzle outlet flow velocity is $V_2 = 5$ m/s to meet this goal. The total water passing area of the nozzles is $S_T = 175.3$ mm$^2$ according to Equation (1). Usually, the number of nozzles in an HQ-size bottom jetting bit is 10, 12, 14, 16, or 18 [24,25]. The structural parameters of the water passage system with different nozzle quantities was determined as shown in Table 1 by solving Equations (2)–(5).

As shown in Table 1, an increase in nozzle quantity causes $S_{\min}(r_1, r_2)$ to increase; that is, the ratio of the total nozzle area to the crown projection area trends downward. This reduces the effective working area of the bit and is not conducive to its service life. Conversely, the grinding ratio $l_k$ decreases as the nozzle quantity increases, which prevents premature scrap of the bit caused by internal diameter deviations. When the number of

nozzles is 10, $S_{\min}$ ($r_1$, $r_2$) is the smallest and $l_k$ is the largest. When the number of nozzles is 18, $S_{\min}$ ($r_1$, $r_2$) is the largest and $l_k$ is the smallest. Taking into account the effects of the bit crown projection area and the grinding ratio on service life, the middle value of $S_{\min}$ ($r_1$, $r_2$) and $l_k$ should be selected when determining the number of nozzles. A 14-nozzle water passage system was designed according to these results.

**Table 1.** Structural parameters of bit water system.

| Nozzle Quantity 2*n* | Main Nozzle Radius $r_1$ (mm) | Width of Main Waterway $b_1$ (mm) | Assistant Nozzle Radius $r_2$ (mm) | Width of Assistant Waterway $b_1$ (mm) | $S_{\min}$ ($r_1$, $r_2$) | $l_k$ |
|---|---|---|---|---|---|---|
| 10 | 3.0 | 8.0 | 3.0 | 8.0 | 0.24 | 1.43 |
| 12 | 2.7 | 7.4 | 2.7 | 7.4 | 0.27 | 1.42 |
| 14 | 2.5 | 7.0 | 2.5 | 7.0 | 0.29 | 1.39 |
| 16 | 2.3 | 6.6 | 2.3 | 6.6 | 0.32 | 1.38 |
| 18 | 2.2 | 6.4 | 2.2 | 6.4 | 0.35 | 1.36 |

## 3. Effects of Bit Water Passage System Parameters on Flow Field at Hole Bottom

### 3.1. Bit Water Passage System Model

The waterway height of the impregnated diamond bit directly affects the cooling effect of drilling fluid on the cutting tooth crown of the bit and, thus, the service life of the bit [26,27]. The waterway height of the bit is usually equal to its working layer height. On the premise of ensuring that the drilling fluid can normally cool the cutting teeth of the bit, the working layer height of the bit should be increased to the greatest extent possible to increase its service life. As per the bit water passage system parameters presented above, a bit model was established in Solidworks as shown in Figure 2. Considering that the waterway height is limited by the cooling effect of the bottom crown of the bit and the manufacturing process, the waterway height is usually less than or equal to 14 mm, so bit models with *h* of 10 mm, 12 mm, and 14 mm were established [28].

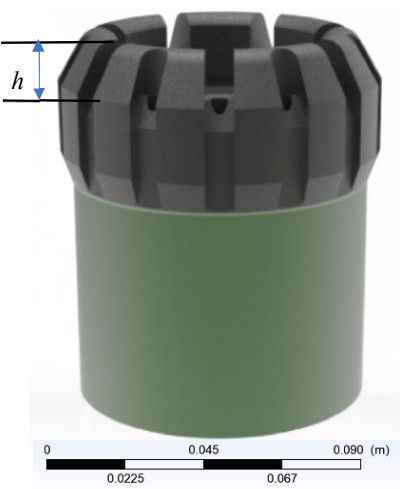

**Figure 2.** Bit model diagram.

The actual assembly of the drilling tool during the drilling process was incorporated into the model as shown in Figure 3. The model encompasses six parts of the water passage: the hole wall, bit, core lifter case, core lifter, stop ring, and core. Solidworks 5 Boolean subtraction operations were used to secure the fluid domain model, one-seventh of which was used as the display object as shown in Figure 4.

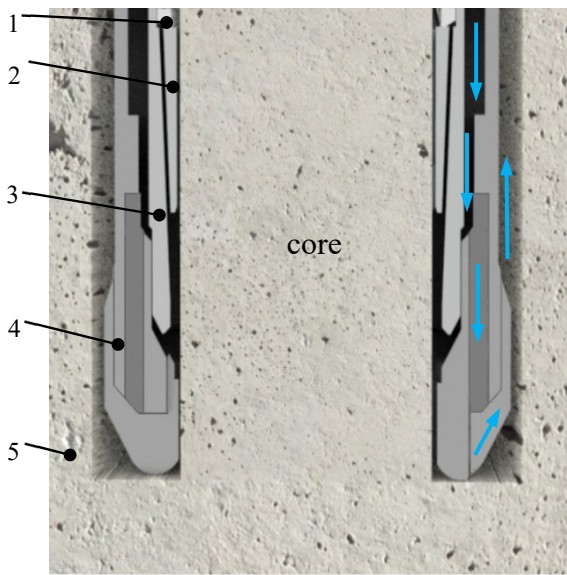

1. Stop ring; 2. Core lifter; 3. Core lifter case;

4. Diamond bit; 5. Hole wall

**Figure 3.** Water passage model diagram.

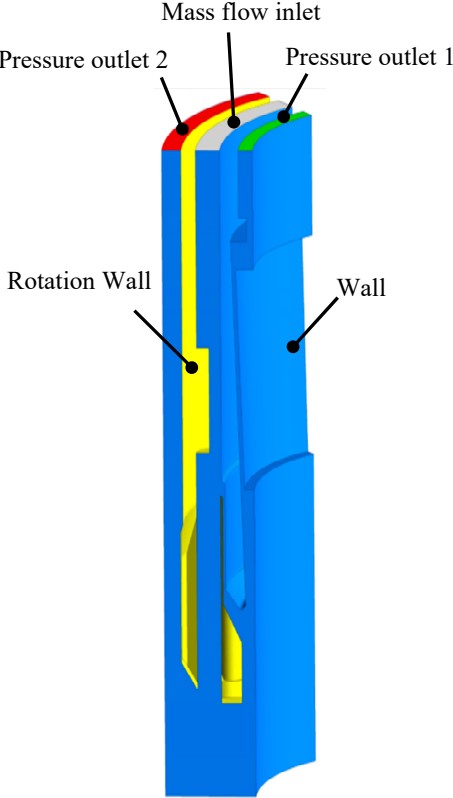

**Figure 4.** Water passage simulation model and boundary diagram.

### 3.2. Simulation Parameters and Boundary Setting

In order to facilitate the setting of the boundary conditions of the internal fluid channel and the axial symmetry of the model, one-seventh of the model is used for simulation. The impact of bit rotation on the flow field at the hole bottom was analyzed as shown in Figure 4, where the rotation boundary of the bit is marked in yellow. The rotation speed is

300 rpm, and the wall boundary is marked in blue. Seawater was used as the drilling fluid medium with a density of 1.03 kg/m$^3$ and pump displacement of 80 L/min. The initial conditions for this simulation include a mass flow inlet and two pressure outlets. The mass flow inlet boundary condition is 0.515 kg/s in the annular clearance between the bit and the core lifter case. Two pressure outlets' boundary is set to 0.89 Mpa. Using the improved mesh quality of fluent, the grid with 5% quality less than 0.5 is increased to more than 0.7, and the grid model diagram is shown in Figure 5.

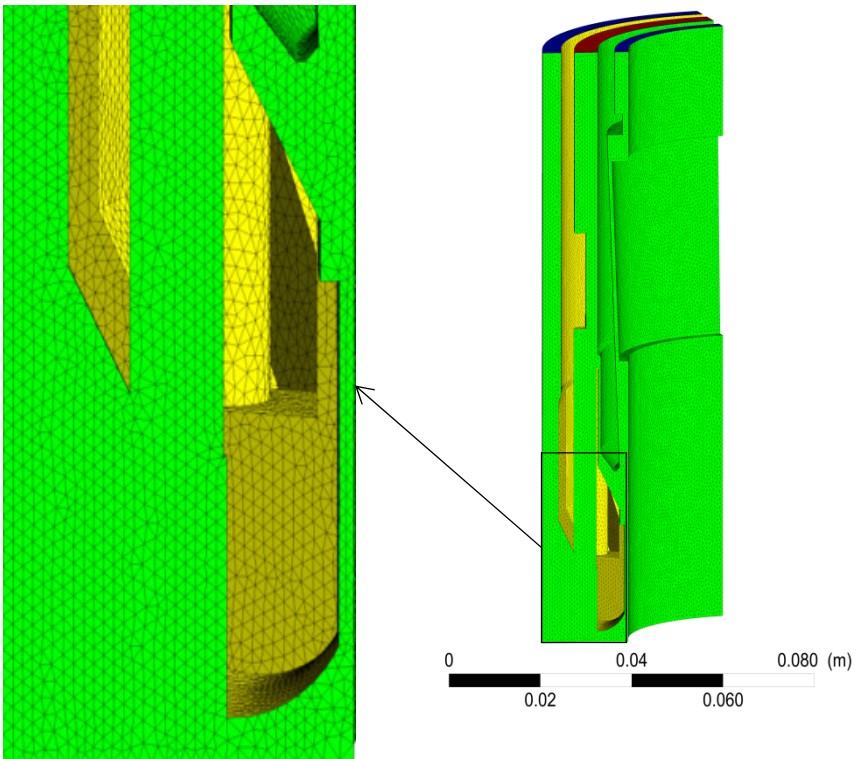

**Figure 5.** Octree algorithm generates grid model diagram.

### 3.3. Effects of Waterway Height on Drilling Fluid Velocity at Bottom Crown of the Bit

Fluid simulations were conducted in Fluent software. The SIMPLE algorithm is used to perform the coupling calculation of pressure and velocity to improve the solution speed, and the convergence accuracy is set to $10^{-3}$. The simulation model assumes that the wall hole and bottom hole are smooth and the seawater is an incompressible Newtonian fluid, ignoring a small number of cuttings at the bottom hole. In the steady-state simulation, considering that the model has many curved walls and the drilling fluid produces spray effects at the hole bottom, the strong swirling flow may be formed in many places. The improved RNG k-epsilon model can better solve the waterway problem of a large degree of curvature and rapid change in water flow rate. Set the time to 180 steps to achieve the required convergence accuracy and take the converged results for analysis. In order to reduce the influence of mesh accuracy on the model, mesh independence verification is carried out. When the total number of meshes reaches 2.3 million, the error of the maximum velocity outlet is less than 5%.

Fluid simulations were conducted in Fluent software. Velocity nephograms of drilling fluid on the bit crown were obtained with *h* values of 10 mm, 12 mm, and 14 mm (Figure 6). The flow velocity of the crown reflects the cooling effects of the drilling fluid and eddy current size on the return of cuttings, making it an important parameter. In the figure, the red coil is the erosion risk area of drilling fluid to the hole wall, the maximum speed of which is 2.9 m/s when *h* = 14 mm. There is a risk of punching the hole wall when drilling in soft strata such as sediments. The black coil in the figure marks the direct injection area

of the nozzles. The bit velocities with different waterway heights in this region range from 2.0 to 2.3 m/s. When *h* is 10 mm or 12 mm, there is little difference in the erosion velocity of the hole wall.

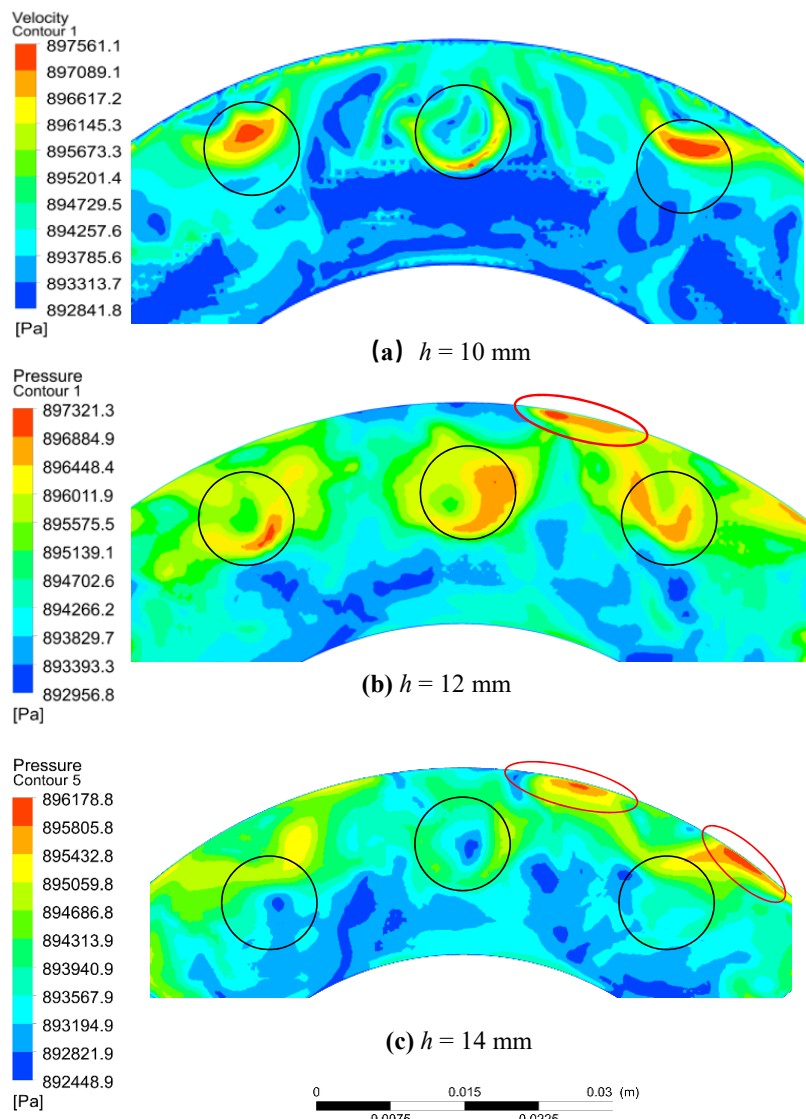

**Figure 6.** Effects of waterway height on crown flow velocity (**a**) *h* = 10 mm; (**b**) *h* = 12 mm; (**c**) *h* = 14 mm.

The drilling fluid sprayed from each waterway converges to form a vortex. The drilling fluid at the hole bottom is in a state of swirling, which is not conducive to the return of cuttings. The swirling state of the drilling fluid at the hole bottom impedes the return of bottom cuttings; however, when the impregnated diamond bit is used for drilling, a small number of cuttings left at the hole bottom can promote diamond exposure there. A small vortex at the bottom of the borehole is, thus, conducive to retaining a small number of cuttings to facilitate diamond exposure. As shown in Figure 6, when *h* is 10 mm, the drilling fluid flow velocity of the larger area on the crown is less than 0.2 m/s, which is not conducive to cooling of the bit crown or the return of carried cuttings. When *h* is 12 mm, there is a small amount of swirl on the crown of the drill bit, and the drilling fluid velocity of the larger area on the crown is greater than 0.7 m/s, which is helpful to the normal return of cuttings. The eddy current on the crown is significant when *h* is 14 mm, which is not conducive to carrying cuttings in the drilling fluid. The bit waterway is optimal at a height of 12 mm.

### 3.4. Effects of Waterway Height on Drilling Fluid Intensity of Pressure

The main waterway section accurately reflects changes in the drilling fluid and its maximum flow velocity in the core, as well as the flow velocity of the hole wall annulus. Figure 7 shows a pressure nephogram of the drilling fluid at the main waterway section. The pressure gradient is large when the drilling fluid reaches nozzle 1 from the inside annular area. When the drilling fluid reaches the bottom "B", the kinetic energy is converted back into pressure energy. Drilling fluid in the bit and the hole wall annulus "W" water flow area decreases, the pressure reaches its lowest value, then the pressure moves back to the outlet. The bottom pressure region area B increases as waterway height $h$ increases, reaching its maximum at $h$ = 14 mm. Without considering the leakage and heat loss at the hole bottom, when $h$ is 10 mm, 12 mm, and 14 mm, the pressure loss is about 17,000 Pa, 17,800 Pa, and 184,000 Pa, respectively. In effect, pressure loss increases as $h$ increases.

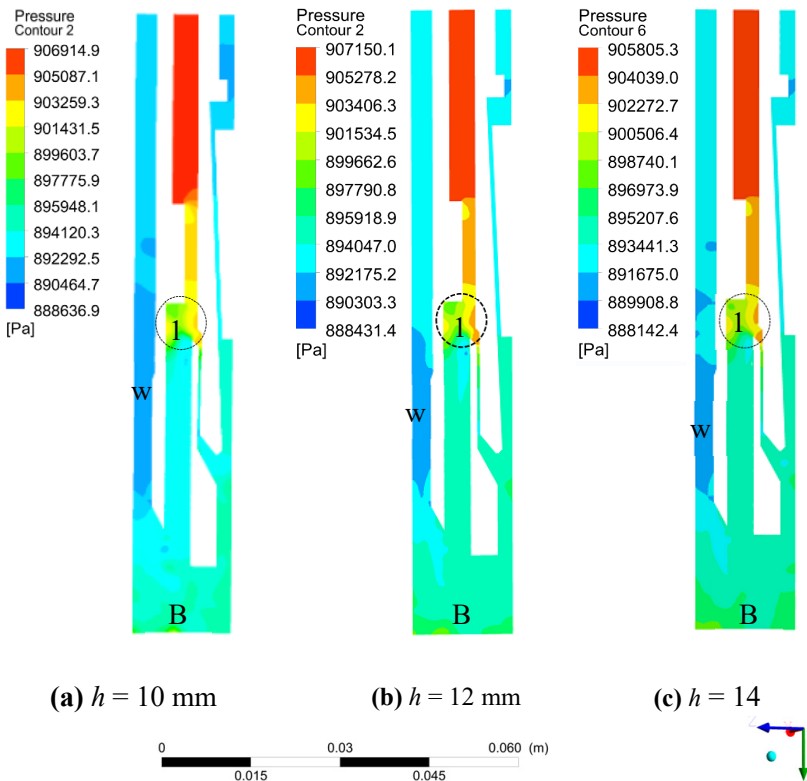

**(a)** $h$ = 10 mm          **(b)** $h$ = 12 mm          **(c)** $h$ = 14

**Figure 7.** Pressure nephogram of main waterway section (**a**) $h$ = 10 mm; (**b**) $h$ = 12 mm; (**c**) $h$ = 14 mm.

### 3.5. Effects of Waterway Height on Drilling Fluid Flow Velocity

Figure 8 shows the velocity profile of the drilling fluid in the main waterway section. As the drilling fluid reaches the nozzles of the bit, the water area decreases, the pressure energy is converted to potential energy, and the flow velocity reaches its maximum. After the drilling fluid reaches the hole bottom, the flow velocity decreases and then returns along the outside annulus of the hole wall. As shown in Figure 8, when the drilling fluid reaches the outer annulus "W", the pressure energy decreases and the flow velocity increases until stabilizing at about 1 m/s. This satisfies the requirements for seafloor core drilling. The area of the black coil at the bottom increases as waterway height increases and its flow pattern grows more disordered, which is not conducive to the return of cuttings.

Region "D" is the direct erosion region of drilling fluid to the core. This area's average velocity and maximum velocity are used to identify the erosion degree of drilling fluid to the core. As shown in Figure 8, the decree of scouring by drilling fluid on D decreases as waterway height increases. The core erosion is largest when $h$ = 10 mm, and the bottom flow field is excessively disordered when $h$ = 14 mm, so the middle height $h$ = 12 mm is optimal.

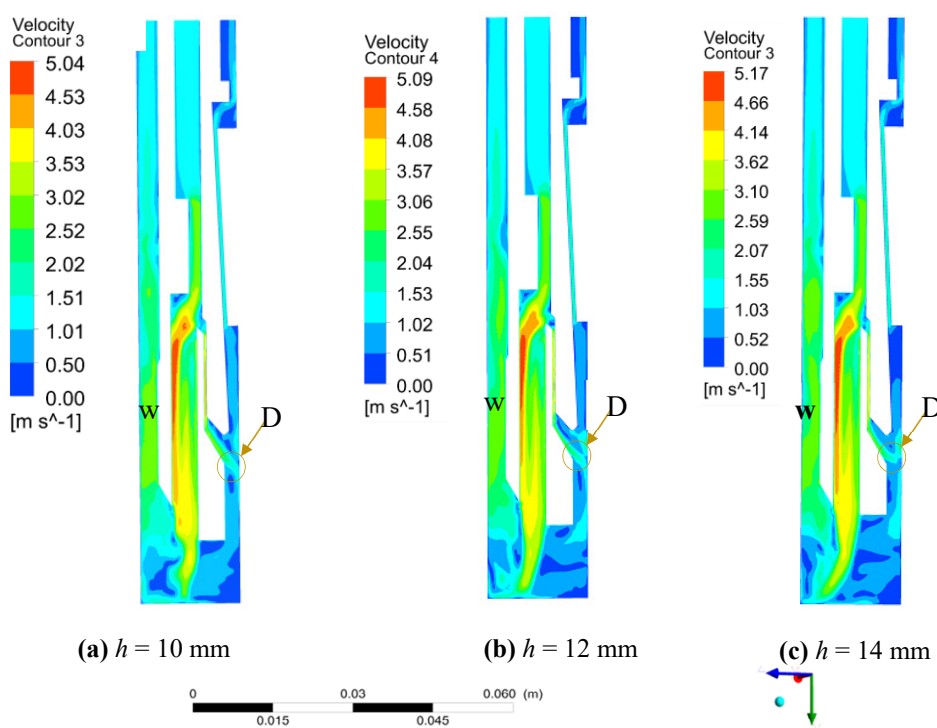

**Figure 8.** Velocity nephogram of main waterway section (**a**) *h* = 10 mm; (**b**) *h* = 12 mm; (**c**) *h* = 14 mm.

## 4. Effects of Drilling Parameters on Bottom Hole Flow Field

### 4.1. Effects of Pump Displacement on Flow Field at Hole Bottom

According to the above simulation, about 90% of the drilling fluid returns from the outside annulus, while the remaining 10% returns along the gaps of the stop ring. Therefore, Equation (6) can be written based on Equation (1).

$$0.9Q = \frac{60\pi V_1}{4}(D_1{}^2 - D_2{}^2) \times 10^{-3} \tag{6}$$

When drilling in soft to medium-hard formations on the seafloor, the return flow velocity of the drilling fluid carrying the cuttings is 0.5–1.4 m/s and the range of the drilling fluid pump displacement *Q* from Equation (6) is 40–110 L/min. According to seafloor core drilling regulations, the rotation speed of the bit is 300 rpm. The flow velocity change rule of the drilling fluid pump volume within the calculated range was obtained through Fluent simulation analysis, as shown in Figure 9. When the pump displacement ranges from 40 to 110 L/min, the average flow velocity of outlet 2 increases from 0.34 m/s to 1.35 m/s and the maximum flow velocity of outlet 2 increases from 0.5 m/s to 1.7 m/s. The maximum flow velocity at outlet 1 also increases from 0.43 m/s to 1 m/s. The drilling fluid pump displacement should be controlled within the range of 50–85 L/min. It can be seen from Figure 10 that with the increase in pump displacement, the up-return velocity of drilling fluid increases linearly. Linear fitting was performed on the data to obtain the regression equation, as shown in Table 2.

As shown in Figure 10, as the pump displacement increases from 40 L/min to 110 L/min, the pressure loss at the hole bottom increases from 5434 Pa to 23,324 Pa. Taking pump displacement as *x* and pressure loss as *y*, the fitted linear equation is *y* = 346.3 *x* − 9577. R-square is 0.997.

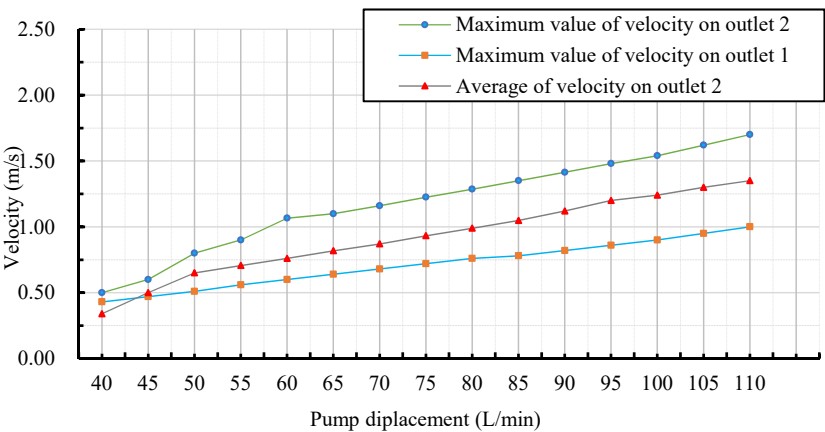

**Figure 9.** Effect of pump displacement on the outlet velocity of the drilling fluid.

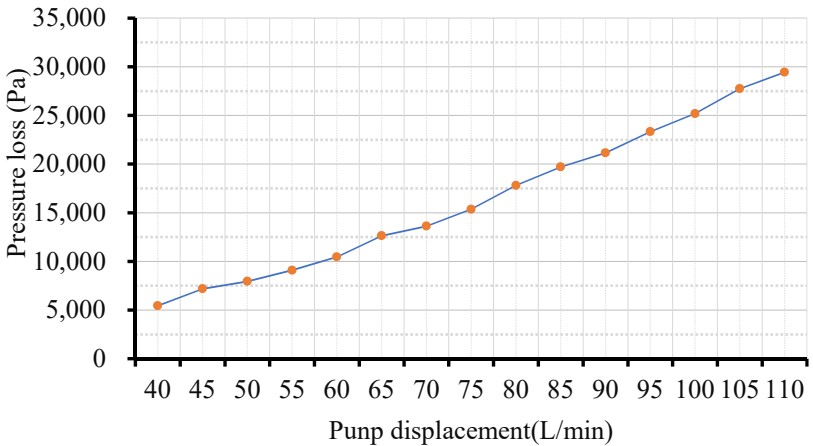

**Figure 10.** Effect of pump displacement on pressure loss of the bottom hole flow field.

**Table 2.** The regression equation for the velocity field of up-return drilling fluid at the bottom hole.

| Velocity Field | Pump Displacement $x$ (L/min) | Equation $y$ (m/s) | R-Square |
|---|---|---|---|
| Maximum value of velocity on outlet 2 | $40 \leq x \leq 60$ | $y = 0.028\,x - 0.57$ | 0.986 |
| | $60 < x \leq 110$ | $y = 0.013\,x + 0.24$ | 0.997 |
| Average of velocity on outlet 2 | $40 \leq x \leq 60$ | $y = 0.025\,x - 0.64$ | 0.943 |
| | $60 < x \leq 110$ | $y = 0.012\,x + 0.022$ | 0.997 |
| Maximum value of velocity on outlet 1 | $40 \leq x \leq 110$ | $y = 0.008\,x + 0.12$ | 0.997 |

### 4.2. Effects of Pump Displacement on Core Surface Velocity

The maximum velocity and average velocity of the core surface reflect disturbance to the core due to drilling fluid scouring. Figure 11 shows the variations in maximum velocity and average velocity of drilling fluid affecting the core surface with pump displacement. As the pump displacement increases from 40 L/min to 110 L/min, the maximum velocity on the core surface increases from 0.7 m/s to 3.2 m/s and the average velocity on the core surface increases from 0.45 m/s to 2.6 m/s. The velocity of drilling fluid on the core surface increases with pump displacement, so the disturbance of drilling fluid on the core intensifies. If the core surface velocity of drilling fluid below 1.5 m/s is the safety value for drilling sediment formation, the pump displacement should be 50–65 L/min. When drilling complex formations, the pump displacement and bit rotation speed can be

appropriately increased to cool the crown and enhance drilling efficiency. It can be seen from Figure 12 that the pump capacity can be divided into three sections for linear fitting of core surface velocity. The linear regression equation is shown in Table 3.

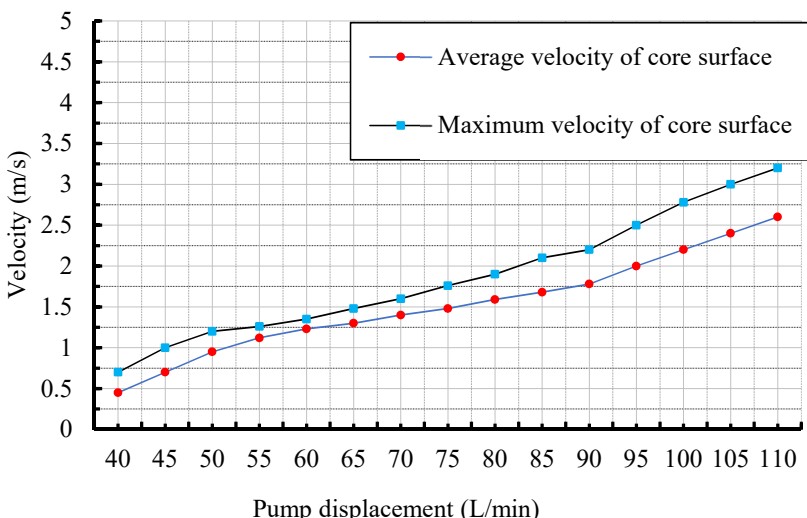

**Figure 11.** Effect of pump displacement on core surface velocity.

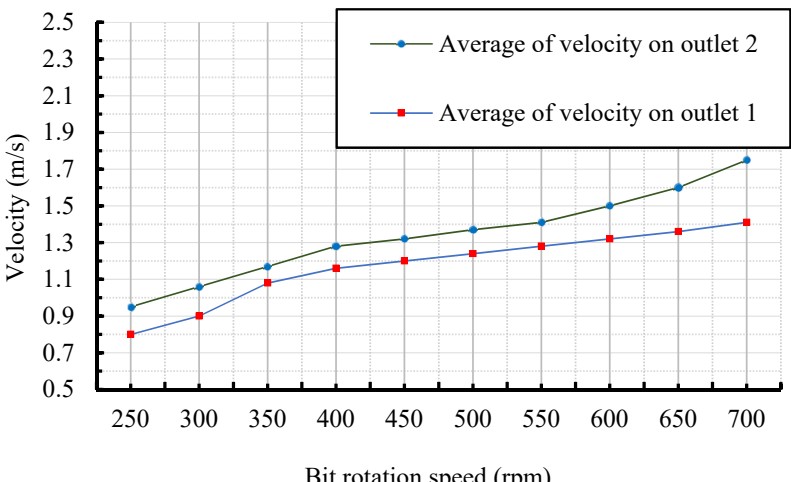

**Figure 12.** Effect of bit rotation speed on back flow velocity of drilling fluid.

**Table 3.** The regression equation of drilling fluid velocity field on the core surface.

| Velocity Field | Pump Displacement $x$ (L/min) | Equation $y$ (m/s) | R-Square |
|---|---|---|---|
| Average velocity of core surface | $40 \leq x \leq 50$ | $y = 0.045\,x - 1.34$ | 0.988 |
| | $50 < x \leq 90$ | $y = 0.018\,x + 0.097$ | 0.998 |
| | $90 < x \leq 110$ | $y = 0.408\,x - 1.884$ | 0.999 |
| Maximum velocity of core surface | $40 \leq x \leq 50$ | $y = 0.037\,x - 0.75$ | 0.943 |
| | $50 < x \leq 90$ | $y = 0.028\,x - 0.32$ | 0.991 |
| | $90 < x \leq 110$ | $y = 0.05\,x - 2.26$ | 0.989 |

*4.3. Effects of Bit Rotation Speed on Drilling Fluid Return Velocity*

According to the seafloor coring drilling regulations, the rotation speed range of the impregnated diamond bit is 250–700 rpm [29]. An appropriate increase in bit rotation

speed can, thus, enhance drilling efficiency. However, such an increase also increases the amount of cuttings and heat that are generated and converts the kinetic energy of the bit to that of the drilling fluid. When the maximum pump displacement of drilling fluid is 80 L/min, the velocity of drilling fluid is as shown in Figure 12. When the bit rotation speed increases from 250 rpm to 700 rpm, outlet 1 increases from 0.8 m/s to 1.41 m/s and outlet 2 from 0.95 m/s to 1.75 m/s. To effectively control the return velocity of drilling fluid, the bit rotation speed should be within the range of 250 rpm to 400 rpm. According to the above data, the linear regression equation is shown in Table 4.

**Table 4.** The regression equation for the velocity field of up-return drilling fluid at the bottom hole.

| Velocity Field | Bit Rotation Speed $x$ (rpm) | Equation $y$ (m/s) | R-Square |
|---|---|---|---|
| Average of velocity on outlet 2 | $250 \leq x \leq 400$ | $y = 0.0022\,x + 0.4$ | 1 |
| | $400 < x \leq 700$ | $y = 0.0016\,x + 0.53$ | 0.933 |
| Average of velocity on outlet 1 | $250 \leq x \leq 400$ | $y = 0.0025\,x + 0.166$ | 0.968 |
| | $400 < x \leq 700$ | $y = 0.0008\,x + 0.825$ | 0.998 |

## 5. Field Application

An impregnated diamond bit was fabricated according to the determined water system parameters as shown in Figure 13. The matrix formula is a WC-based matrix formula system, which has good hardness and bending strength and is conducive to the service life of the bit [30]. Sediments mainly dominate the strata within 200 m of the seafloor, so the coarse-grained and low-concentration diamond parameter scheme was selected. This scheme was expected to improve the rock fragmentation efficiency of the bit by increasing the exposed height of the single-grained diamond and the specific pressure of the bit crown. The hard, brittle SiC particles were added to the matrix to promote diamond exposure at a concentration of 15% and an average particle size of 425 μm [13]. The cutting tooth structure is conical, which improves the bit crown pressure and drilling stability. The parameters of the diamond bit are listed in Table 5.

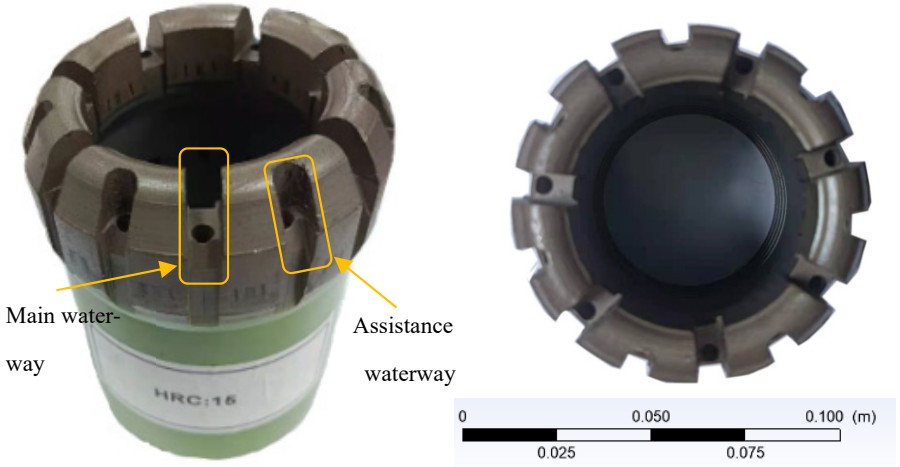

Main water-way

Assistance waterway

HRC:15

| 0 | 0.050 | 0.100 (m) |
| 0.025 | 0.075 | |

**Figure 13.** Finished diamond bit products.

**Table 5.** Diamond bit parameters.

| Bit Size (mm) | Matrix Hardness (HRC) | Diamond Concentration (vol%) | Diamond Size (mesh) |
|---|---|---|---|
| 96/62 | 15 | 55% | 30/35 |

Field drilling tests were conducted in a mineral resource exploration project in a sea area of China. The seabed sediments in the target area are mainly composed of sediments and unconsolidated flow sand with hard, thin flint layers. A seafloor multi-purpose drilling rig with wire-line coring drilling technology was operated throughout the test. The height of the rig is 5.6 m and the bottom size is $2.2 \times 2.2$ m$^2$. The total weight of the seafloor drill is 8.3 tons in air and 6.7 tons in water. The maximum operating water depth is 3500 m. The seafloor drill can carry 25 core pipes simultaneously and has a maximum drilling depth of 62.5 m. Five stations drill over an operating water depth range of 900–1200 m, single hole drilling depth of 62.5 m, and drilling diameter of 97 mm. Seawater was used as drilling fluid with drilling pressure of 8–15 kN, rotation speed of 250–400 rpm, drilling fluid pump displacement of 50–80 L/min, and it ran to a depth of 2.5 m in each iteration. The test is shown in Figure 14 and information regarding drilling operations is given in Table 6.

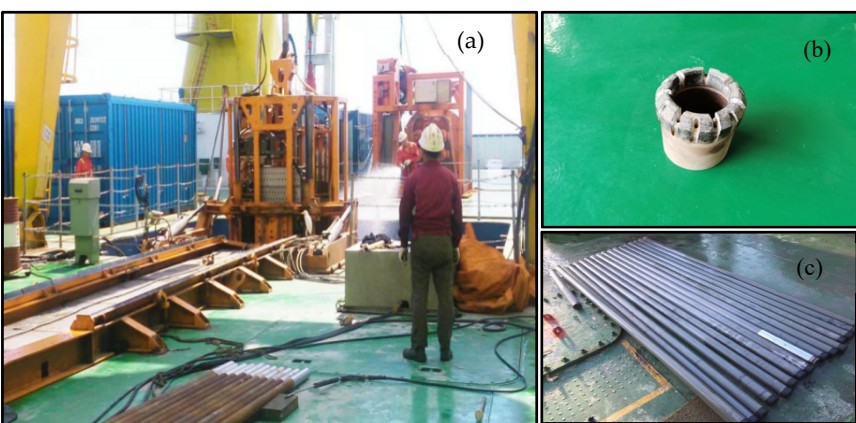

**Figure 14.** Field application of diamond bit in seafloor coring drilling, (**a**) seafloor drill deployment operation; (**b**) used diamond bits; (**c**) core samples.

**Table 6.** Drilling operation.

| Station Number | Drilling Depth (m) | Average Drilling Efficiency (m/h) | Coring Recovery Rate (%) |
|:---:|:---:|:---:|:---:|
| 1 | 62.5 | 2.8 | 86.5 |
| 2 | 62.5 | 3.1 | 87.6 |
| 3 | 62.5 | 3.4 | 83.2 |
| 4 | 62.5 | 2.9 | 90.4 |
| 5 | 62.5 | 3.3 | 85.4 |

The total cumulative drilling reached 312.5 m and average drilling efficiency reached 3 m/h. The average coring recovery rate was 86.6%. In the drilling process, the bit did not appear to be burnt due to a blockage of the water passage system. This suggests that the bit water passage system design is reasonable and can satisfy the real-world requirements for adaptive seafloor drill switching. To this effect, the proposed design may provide a workable reference for water passage systems and drill bit parameter schemes.

## 6. Conclusions

Seafloor drill operating and formation characteristics were combined in this study to design a water passage system suitable for bottom jetting diamond bits based on the basic theory of multi-objective optimization. Based on fluid dynamics theory and considering the effects of bit rotation on the flow field at the hole bottom, the influence law of the water passage system parameters and drilling parameters of HQ-size bits on the flow field at the hole bottom was determined. The conclusions of this work can be summarized as follows.

(1) Setting the optimization goal of the water passage system as the maximum projection area of the cutting tooth can improve the normal service life of the bit, as grinding length ratio can create lopsided wear between inner and outer diameters.

(2) The flow field of drilling fluid at the hole bottom grows increasingly disordered as the bit waterway height increases, which drives down the efficiency of carrying cuttings and increases pressure loss at the hole bottom.

(3) The optimal bit rotation speed is 250–400 rpm. When drilling in conventional formations, pump displacement control in the range of 50–80 L/min is optimal. When drilling in sediment formations, pump displacement control in the range of 50–65 L/min can reduce drilling fluid damage to the core.

**Author Contributions:** J.W. expounds on the theory on which the paper is based, makes engineering tests, and makes final modifications to the paper; D.Q. has carried on the thesis writing and the numerical simulation; Y.S. establishes a simulation model; F.P. has guided the numerical simulation of the paper. All authors have read and agreed to the published version of the manuscript.

**Funding:** This research is supported by the National Natural Science Foundation of China (No. 41702390), Open Research Fund Program of Key Laboratory of Metallogenic Prediction of Nonferrous Metals and Geological Environment Monitoring (Central South University), Ministry of Education (2019YSJS15), National Key R&D Program of China (No. 2017YFC0307501) and Hunan Provincial Natural Science Foundation China (No. 2018JJ3173).

**Institutional Review Board Statement:** Not applicable.

**Informed Consent Statement:** Not applicable.

**Data Availability Statement:** The study did not report any data.

**Acknowledgments:** All authors thank the anonymous reviewers for constructive comments that helped improve this manuscript.

**Conflicts of Interest:** The authors declare no conflict of interest.

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
