# Peer review of "Design of Diamond Bits Water Passage System and Simulation of Bottom Hole Fluid Are Applied to Seafloor Drill"

_jmse, doi:10.3390/jmse9101100_

Round 1
Reviewer 1 Report
The article by the authors is devoted to an important and topical issue related to the analysis of the process of drilling on the seabed. An important aspect of the work, in my opinion, is the combination of drilling performance and reservoir characteristics using the theory of multi-criteria optimization. The studies presented in the work are undoubtedly of interest to readers in the area under consideration.
However, it would be necessary to clarify a number of comments that are available to the article:
- The article should have dwelt in more detail on the analysis of Figure 8, which reflects the speed of the influence of the height of the waterway on the current speed and explain exactly how the results were obtained using the Fluent software.
- From the analysis of Figure 9, it is not entirely clear how the stabilization rate of about 1 m / s was determined (page 9) and whether a more significant change in the range of recorded parameters is possible for various conditions presented in Figures 8, 9, 10.
- The results presented in Figures 11, 12, 13 and 14 are of great interest. Regression equations could be presented, with the help of which it would be possible to carry out short-term and long-term forecasting of the studied parameters.
- It is not entirely clear from the article, at which specific production facilities the results were obtained or only in laboratory conditions? Are there any patents on the results obtained?
- In the article, it was possible to strengthen the list of references with a number of studies by Russian and American scientists, as well as remove formulas (1), (2), making references to them from well-known literary sources.
Reviewer 2 Report
- Please consider reviewing the abstract and highlight the novelty, major findings, and conclusions.
- Just before the last paragraph of the abstract, the authors are encouraged to answer the following question: What is the research gap did you find from the previous researchers in your field? Mention it properly. It will improve the strength of the article.
- Regarding figures 1 and 2, I am not sure if it is useful to add this kind of information in a scientific paper, this kind of knowledge is more suitable for books or student thesis chapters.
- Please add a list of nomenclature for all the symbols and letters used in the study at the end of the manuscript (recommended).
- Figure 4 please add a scale bar.
- The authors should not add two different figures in the same place (for example figures 5 and 6) either rename them as (a) and (b) or add each figure separate from each other.
- The title of the manuscript doesn’t reflect that there is an FE model performed in the study, please consider revising the title of the manuscript to reflect that and better reflect what was done in this work.
- What are the limitations of the FE study? How can the boundary conditions affect the results of the FE study?
- Figure 9 add (a) (b) and (c), same for figure 15 and add a scale bar.
- Figure 16 add (a) (b) and (c) also use a more appropriate caption for this figure, just saying field test is not sufficient.
- Line 330 “An appropriate proportion of hard, brittle SiC particles” it is not clear to me what does the authors mean by an appropriate proportion! According to what? Industry recommendation, trial and error or based on something else?
- Line 325 “which has good hardness” please use numbers to describe such things, what is the value of good hardness, and it is good in comparison to what? Please support all your claims with references if possible.
- In all the graphs I noticed that the trends are always linear, any explanation for that?
- The authors are encouraged to include more detailed discussion and critically discuss the observations from this investigation with existing literature.
Round 2
Reviewer 2 Report
All questions were answered and paper can be accepted fpr publications, congratulations.